# MEMS Gyroscope Temperature Compensation Based on Improved Complete Ensemble Empirical Mode Decomposition and Optimized Extreme Learning Machine

**DOI:** 10.3390/mi15050609

**Published:** 2024-04-30

**Authors:** Zhihao Zhang, Jintao Zhang, Xiaohan Zhu, Yanchao Ren, Jingfeng Yu, Huiliang Cao

**Affiliations:** 1Tsinghua Shenzhen International Graduate School, Tsinghua University, Shenzhen 518055, China; 18890037230@163.com (Z.Z.);; 2College of Mechanical and Electrical Engineering, Central South University, Changsha 410083, China; 3Quanzhou Yunjian Measurement Control and Perception Technology Innovation Research Institute, Quanzhou 362000, Chinayujingfeng@yjqz.org (J.Y.); 4Key Laboratory of Instrumentation Science & Dynamic Measurement, Ministry of Education, North University of China, Taiyuan 030051, China

**Keywords:** dual-mass MEMS gyroscope, temperature compensation, improved complete ensemble empirical mode decomposition with adaptive noise, sample entropy, time–frequency peak filtering, non-dominated sorting genetic algorithm-II, extreme learning machine

## Abstract

Herein, we investigate the temperature compensation for a dual-mass MEMS gyroscope. After introducing and simulating the dual-mass MEMS gyroscope’s working modes, we propose a hybrid algorithm for temperature compensation relying on improved complete ensemble empirical mode decomposition with adaptive noise (ICEEMDAN), sample entropy, time–frequency peak filtering, non-dominated sorting genetic algorithm-II (NSGA II) and extreme learning machine. Firstly, we use ICEEMDAN to decompose the gyroscope’s output signal, and then we use sample entropy to classify the decomposed signals. For noise segments and mixed segments with different levels of noise, we use time–frequency peak filtering with different window lengths to achieve a trade-off between noise removal and signal retention. For the feature segment with temperature drift, we build a compensation model using extreme learning machine. To improve the compensation accuracy, NSGA II is used to optimize extreme learning machine, with the prediction error and the 2-norm of the output-layer connection weight as the optimization objectives. Enormous simulation experiments prove the excellent performance of our proposed scheme, which can achieve trade-offs in signal decomposition, classification, denoising and compensation. The improvement in the compensated gyroscope’s output signal is analyzed based on Allen variance; its angle random walk is decreased from 0.531076°/h/√Hz to 6.65894 × 10^−3^°/h/√Hz and its bias stability is decreased from 32.7364°/h to 0.259247°/h.

## 1. Introduction

As a very important micro inertial device, the MEMS gyroscope is widely used in aerospace, intelligent electronics, virtual reality and other fields [1,2,3,4]. It can measure the angular velocity of objects and provide data support for navigation, attitude control and other applications. However, a temperature change in the internal thermal-sensitive material leads to the temperature drift phenomenon, which affects the stability and accuracy of its output signal [5]. Temperature compensation can offset the effect of temperature on the MEMS gyroscope and can improve its accuracy and long-term stability [6]. The research and development of temperature compensation methods is one of the current research hotspots for MEMS gyroscopes.

The compensation methods for MEMS gyroscopes are broadly categorized into two approaches: hardware compensation and software compensation. These methods aim to mitigate the detrimental impacts of temperature fluctuations on gyroscope performance, ultimately enhancing the gyroscope’s accuracy and stability across diverse operational scenarios. Hardware compensation methods primarily entail the design and integration of physical structures or components within the gyroscope system to counterbalance temperature-induced errors. This approach often includes embedding temperature sensors directly into the gyroscope package to monitor ambient temperature changes. Through measuring these fluctuations, the sensors offer feedback that can be used for adjusting the gyroscope’s operational parameters, such as deviation and scale factors, in real time to counteract temperature effects [7]. Furthermore, innovative structural designs, such as the design of an I-shaped bulk acoustic resonator, are leveraged to enhance temperature stability and enable dynamic frequency adjustments across a broad spectrum [8]. Tao et al. explored a dual closed-loop control strategy featuring differential modulation for resonant integrated optic gyroscopes, enhancing detection precision and minimizing noise interference, which significantly improves gyroscope reliability and reduces control inaccuracies across varying conditions [9]. Reference [10] introduced a MEMS gyroscope with a unique design enhancing resilience to manufacturing and environmental changes. It employs a single-DOF drive and a two-DOF sensing mechanism, enabling independent frequency and bandwidth adjustments, thus mitigating previous limitations in gyroscopic robustness. In [11], a real-time temperature self-sensing hardware compensation algorithm was designed for high-*Q* MEMS gyroscopes, which greatly improves the measurement accuracy of gyroscopes. Although hardware compensation methods are more accurate and reliable, such methods have a long development cycle, have limited adaptability, and are not easy to promote. On the other hand, the software compensation method based on the use of various kinds of artificial intelligence algorithms to learn the complex temperature–error relationship and provide adaptive compensation solutions has high flexibility and adaptability and can be easily realized and optimized through algorithm adjustment [12]. The gyroscope’s output is often mixed with noise and temperature drift, which need to be dealt with separately. The software compensation algorithms can be divided into two architecture types: serial and parallel. In a serial architecture, the signal is first denoised, and then temperature compensation is performed, which leads to the loss of useful signals. On the contrary, a parallel architecture decomposes the signal first and then selects the corresponding processing means for different signals [13]. Software compensation algorithms usually rely on adaptive signal decomposition methods, denoising methods, neural networks, etc.

The parallel processing architecture carries out noise reduction and temperature compensation at the same time. The most commonly used signal decomposition algorithms include empirical mode decomposition [14], local mean decomposition [15] and variational mode decomposition [16]. Among them, empirical mode decomposition and local mean decomposition suffer from the interference of mode aliases, while variational mode decomposition relies on the researcher’s experience in setting reasonable decomposition parameters, which have a great impact on the decomposition results. Based on empirical mode decomposition, researchers have introduced advanced techniques such as ensemble empirical mode decomposition [17] and complete ensemble empirical mode decomposition with adaptive noise [18], which integrate noise to facilitate decomposition and alleviate modal mixing. Furthermore, Colominas et al. proposed a refined approach known as improved complete ensemble empirical mode decomposition with adaptive noise (ICEEMDAN). Through the incorporation of specialized white noise during decomposition, experiments have demonstrated that ICEEMDAN exhibits superior decomposition performance and versatility, effectively mitigating mode aliases [19]. Consequently, ICEEMDAN is employed in this study to decompose the gyroscope output signal for subsequent analysis. The mainstream denoising algorithms include wavelet transform, Kalman filtering, Fourier transform, etc., all of which have different degrees of drawbacks. Fourier transform lacks time-domain localization ability and is not suitable for non-stationary signals. Kalman filtering involves matrix operations, leading to high computational complexity and significant signal distortion. Although the wavelet threshold denoising algorithm proposed by Johnstone et al. has excellent denoising performance, the selection of parameters is challenging [20,21]. Based on instantaneous frequency estimation of the Wigner–Ville distribution, time–frequency peak filtering was proposed by Boashash et al. It features high time–frequency positioning accuracy, high computational efficiency, and strong noise suppression. In addition, its denoising capability can be adjusted by modifying the window length [22]. In addition, high-precision compensation models are key in parallel temperature compensation algorithms, and various artificial intelligence technologies can establish prediction models between the temperature and the gyroscope’s output. Shen et al. fit a compensation model for temperature drift based on a genetic-algorithm-optimized Elman neural network [23]. Song et al. introduced a novel approach that merges the artificial fish swarm algorithm with a back-propagation neural network, aiming to enhance the accuracy of fiber optic gyroscopes’ output [24]. Li et al. proposed a novel temperature error model based on a radial basis function neural network, improved with particle swarm optimization and regularization methods, to enhance the accuracy and environmental adaptability of the ring laser gyroscope [25]. In addition, support vector regression, wavelet neural networks, fuzzy control theory and so on are often used to build predictive models. Among them, extreme learning machine is widely used because of its simple structure and superior generalization ability [26,27,28]. Since the input weights of extreme learning machine are generated randomly, the instability of the output matrix of the hidden layer results in large connection weights of the output layer, thus increasing the structural risk [29,30].

Based on the above analysis, this study examines the temperature compensation problem for a dual-mass MEMS gyroscope. We introduce the working principle and working mode of the gyroscope, and we formulate a temperature compensation experiment. To solve the mixing problem of noise and temperature drift in the gyroscope’s output signal, we propose an innovative parallel temperature compensation and noise reduction scheme. Firstly, the output signal of the gyroscope is decomposed by improved complete ensemble empirical mode decomposition with adaptive noise, which has a high identification rate. Then, the sample entropy (SE) is used to classify a series of intrinsic mode functions, which can be divided into the noise segment, mixed segment and feature segment. For the noise segment and the mixed segment with different noise levels, the time–frequency peak filtering (TFPF) method with different window lengths is used to achieve a compromise between noise removal and signal retention. For the feature segment with temperature drift, we use non-dominated sorting genetic algorithm-II (NSGA II) to optimize extreme learning machine (ELM) with the prediction error and the 2-norm of the output-layer connection weight as optimization objectives to establish a high-precision temperature compensation model. Finally, the processed components are reconstructed to obtain the final output signal. Through comparative analysis of the experimental results, we prove that the proposed algorithm has excellent performance and effect in gyroscope temperature compensation.

The rest of this paper is organized as follows: Section 2 introduces the working principle of our developed dual-mass MEMS gyroscope. Our proposed algorithm is presented in Section 3. In Section 4, the temperature experiment and algorithm verification are described. Section 5 presents the conclusions.

## 2. Introduction of the Dual-Mass MEMS Gyroscope

In this article, a dual-mass MEMS gyroscope [5] is introduced and utilized for experiments, as illustrated in Figure 1. The gyroscope operates in drive mode and sense mode. The drive mode consists of components such as a driving comb and a driving spring, facilitating movement along the X-axis. Conversely, the sense mode, comprising the sense comb and sense spring, enables movement along the Y-axis. The masses shown in Figure 1 oscillate along the negative direction of the X-axis in both modes. Additionally, to prevent displacement coupling, the two modes of the gyroscope are intentionally isolated. When angular velocity Ωz is applied around the Z-axis, the resulting Coriolis force from the vibrating mass is transmitted along the Y-axis to the frame and monitored through an electrical circuit.

The gyroscope functions on the tuning fork principle, featuring a U-type connecting spring linking the dual-drive mass block and a drive spring connecting the dual-sense mass block. To examine its operational modes, Ansys software (Ansys 2022 R1) was employed for the simulation, as depicted in Figure 2. Notably, there is a significant frequency gap of over 1000 Hz between the first and fourth operational modes, with the fourth mode exhibiting a quality factor exceeding 2000, identifying it as the gyroscope’s drive mode. Figure 2a–d sequentially present the simulation results for the gyroscope’s operational modes. The first mode, the drive in-phase mode, involves the gyroscope’s double mass vibrating consistently with the X-axis. Following this, the second mode, the sensing in-phase mode, sees both masses vibrating simultaneously along the Y-axis direction. In contrast, the third mode, the sensing anti-phase mode, exhibits oscillations in the Y-axis contrary to the direction of the two mass blocks. Lastly, the fourth mode, the drive anti-phase mode, demonstrates oscillations of the gyroscope’s two mass blocks counter to the Y-axis. Additionally, the resonant frequencies for these four modes are observed at 2623 Hz, 3342 Hz, 3468 Hz and 3484 Hz. Remarkably, the fourth mode acts not only as the drive anti-phase mode but also as the primary driving mode. From this analysis, it is evident that the gyroscope’s two mass blocks have two degrees of freedom, while the drive mode and frame each exhibit only a single degree of freedom.

In Figure 3, the drive comb measures the displacement of the drive frame *x*(*t*) using a split amplifier. To achieve the required phase alignment with the signal *V_dac_Sin*(*ω_d_t*), a 90° phase delay is applied to the signal. Subsequently, *V_dac_* is extracted via a full-wave rectifier and low-pass filter and concurrently compared to the reference voltage *V_ref._* A control signal is generated when the comparator’s output passes through the integrator controller. This control signal drives the DC signal *V_DC_* to accumulate into *V_dac_Sin*(*ω_d_t*), thereby exciting the drive mode. The left and right sensitive mass blocks’ motion signals are captured by a differential detection amplifier. Afterward, a second differential amplifier processes the output signal to produce the total sense motion signal, *V_stotal_*. This signal is then demodulated with *V_dac_Sin*(*ω_d_t*) and subsequently filtered using a low-pass filter to obtain the ultimate sensitive motion signal.

## 3. Algorithms and Models

### 3.1. Improved Complete Ensemble Empirical Mode Decomposition with Adaptive Noise

EMD is a classical algorithm for complex signal analysis and processing. EMD works by decomposing the signal into some amplitude–frequency modulation functions and one monotonic trend signal. Although EMD has good effects in the analysis of complex signals, it leads to the phenomenon of mode mixing due to its local characteristics. To solve this challenge, EEMD was proposed to mitigate the mode mixing phenomenon by incorporating noise into the decomposition process. Subsequently, CEEMDAN based on adaptive noise was proposed, which is an important improvement on EEMD and solves the influence of auxiliary noise on the mode number. However, CEEMDAN still has deficiencies compared to EEMD, such as residual noise in the modes and signal characteristics appearing after the false mode. The above problems were solved through ICEEMDAN. Different from the previous methods, which add Gaussian white noise in the auxiliary decomposition, ICEEMDAN uses special white noise *E_k_*[*w*^(*i*)^] in the auxiliary decomposition; the noise originates from the *k*th component of white Gaussian noise, which undergoes decomposition via EMD. A brief introduction of ICEEMDAN follows [31,32].

Before describing the algorithm steps, several operation symbols are first defined. *M*(.) is the operator that generates the local mean value, and the *k*th IMF value after EMD decomposition can be obtained through the *E_k_*(.) operator. Obviously, *E*_1_(*x*) = *x* − *M*(*x*).

The decomposition process proceeds as follows:

Step 1. Adding certain noise. In Equation (1), *w*^(*i*)^ represents the *i*th added noise, and *β*_0_ is the standard deviation of adding noise.
(1)$$\left\{\begin{array}{c} x^{\left(1\right)}=x+\beta_{0}E_{1}\left[w^{\left(1\right)}\right]\\ x^{\left(2\right)}=x+\beta_{0}E_{1}\left[w^{\left(2\right)}\right]\\ \vdots \\ x^{\left(i\right)}=x+\beta_{0}E_{1}\left[w^{\left(i\right)}\right] \end{array}\right..$$

Step 2. EMD calculates the local mean for each *x*^(*i*)^, and the mean of their local means is taken as the first residual *r*_1_; then, we can obtain the first IMF value, *c*_1_:(2)$$\left\{\begin{array}{l} r_{1}=\frac{1}{I}\overset{I}{\underset{i=1}{\sum }}M\left[x^{\left(i\right)}\right]\\ c_{1}=x-r_{1} \end{array}\right..$$

Step 3. The second IMF value, *c*_2_, is calculated immediately:(3)$$\left\{\begin{array}{l} c_{2}=r_{1}-r_{2}\\ r_{2}=\frac{1}{I}\overset{I}{\underset{i=1}{\sum }}M\left\{r_{1}+\beta_{1}E_{2}\left[x^{\left(i\right)}\right]\right\} \end{array}\right..$$

Step 4. Similarly, *k* IMF values are calculated according to the above steps, where
(4)$$\left\{\begin{array}{l} c_{k}=r_{k-1}-r_{k},k=2,3,\ldots ,N\\ r_{k}=\frac{1}{I}\overset{I}{\underset{i=1}{\sum }}M\left\{r_{k-1}+\beta_{k-1}E_{k}\left[x^{\left(i\right)}\right]\right\} \end{array}\right..$$

After obtaining *k* modes and adding the residuals, the original signal can be rewritten as
(5)$$x=\overset{k}{\underset{i=1}{\sum }}c_{k}+r_{k}.$$

ICEEMDAN is used to decompose the gyroscope’s output into a series of intrinsic mode functions (IMFs). Our proposed hybrid algorithm needs to classify the obtained IMFs to identify different components; thus, it uses SE as a decision rule to distinguish these IMFs.

### 3.2. Sample Entropy

SE is an effective measurement method that can measure the regularity and complexity of complex signals by calculating the possibility of generating different patterns in the sequence [33]. The SE is not dependent on the data length and has high detection accuracy. A stable estimate can be obtained with a relatively short data sequence. The principle of sample entropy is as follows:

Step. 1 Suppose that time series *h* is composed of *n* data, that is, $h=\{h\left(1\right),h\left(2\right),\ldots ,$ $h\left(n\right)\}$, and then construct a m-dimensional vector based on the original signal, which is $H_{m}\left(1\right),\cdots ,H_{m}(n-m+1)$, where $H_{m}\left(i\right)$ = $\left\{h\left(i\right),h\left(i+1\right),\cdots ,h\left(i+m-1\right)\right\}$, 1 ≤ *i* ≤ *n* − *m* + 1.

Step. 2 The distance *d_ij_* between $H_{m}\left(i\right)$ and $H_{m}\left(j\right)$ is
(6)$$d_{ij}=d\left[h\left(i\right),h\left(j\right)\right]=\underset{l=0,\ldots ,m-1}{\max}\left|h\left(i+l\right)-h\left(j+l\right)\right|,$$
where *j* = 1, 2, …, *n* − *m*, and *j* ≠ *i*.

Step. 3 For a given $H_{m}\left(i\right)$, compute the distances between $H_{m}\left(i\right)$ and $H_{m}\left(j\right)$ and calculate the number $\;C_{i}$ of them that are less than or equal to *p*:(7)$$C_{i}^{m}\left(p\right)=\frac{1}{n-m-1}C_{i}.$$

Step. 4 $C^{m}(p)$ is defined as
(8)$$C^{m}\left(p\right)=\frac{\sum_{i=1}^{n-m}C_{i}^{m}\left(p\right)}{n-m+1}.$$

Step. 5 When the dimension of the sequence is increased from m to *m* + 1, compute the distances between $H_{m+1}\left(i\right)$ and $H_{m+1}\left(j\right)$ and count the number${\;D}_{i}$ of them that are less than or equal to *p*:(9)$$D_{i}^{m}\left(p\right)=\frac{1}{n-m+1}D_{i}.$$

Step. 6 Then, $D^{\left(m\right)}(p)$ can be obtained:(10)$$D^{m}\left(p\right)=\frac{\sum_{i=1}^{n-m}D_{i}^{m}\left(p\right)}{n-m+1}.$$

In the above definition, under the condition of similar tolerance *p*, $C^{m}(p)$ and $D^{\left(m\right)}(p)$ denote the probability of two sequences matching either *m* or *m* + 1 points individually. Subsequently, SE is defined as
(11)$$SE\left(m,p\right)=-\underset{N\to \infty }{\lim}\left\{\ln D^{m}\left(p\right)/C^{m}\left(p\right)\right\}.$$

Equation (11) can be rewritten into Equation (12) if n is a finite number.
(12)$$SE\left(m,p\right)=-\ln\left[\frac{D^{m}\left(p\right)}{C^{m}\left(p\right)}\right].$$

The smaller the value of SE, the higher the similarity of the signal sequence, which means that the sequence is regular and can be considered a useful signal, and vice versa.

### 3.3. Time–Frequency Peak Filtering

TFPF stands as a pivotal noise elimination technology. Renowned for its prowess in extracting meaningful signals amidst noisy environments, it finds extensive applications across various engineering domains. Operating primarily on the principles of WVD and instantaneous frequency estimation theory, the TFPF algorithm serves to filter and denoise signals. Despite the widespread utilization of WVD in engineering, owing to its commendable time–frequency focusing capabilities, its efficacy diminishes when processing multi-component signals due to the emergence of cross terms, which compromise its time–frequency resolution. To address the challenges posed by cross terms in TFPF, we integrate the pseudo-Wiener–Ville distribution (PWVD). Essentially, TFPF involves encoding the noisy signal to convert it into an analytic signal of the instantaneous frequency, which enables an estimation of the effective signal through instantaneous frequency estimation. Let us consider a signal *y*(*t*) contaminated with noise:(13)$$y\left(t\right)=x\left(t\right)+n\left(t\right),$$
where *x*(*t*) represents the useful signal in *y*(*t*) and *n*(*t*) represents the noise in *y*(*t*). The steps for denoising *y*(*t*) by TFPF are as follows [34]:

Step 1. By subjecting the signal *y*(*t*), which incorporates noise, to frequency modulation, the analytic signal *z*(*t*) is derived:(14)$$z\left(t\right)=e^{j2\pi \mu \int_{0}^{t}y\left(\lambda \right)d\lambda },$$
where *µ* is the frequency modulation index.

Step 2. The spectrum of the PWVD for *z*(*t*) is
(15)$$PW_{2}\left(t,f\right)=\int_{-\infty }^{\infty }h\left(\tau \right)z\left(t+\frac{\tau }{2}\right)z^{*}\left(t-\frac{\tau }{2}\right)e^{-j2\pi ft}d\tau .$$
where *t* represents time, *τ* is the integral variable, *f* denotes frequency, z* represents the conjugate operator of *Z* and *h*(*τ*) is the window function.

Step 3. We compute the peak value of the PWVD spectrum of the analytic signal to estimate its instantaneous frequency based on the maximum likelihood estimation principle. Consequently, the amplitude of the original effective signal is estimated:(16)$$f_{z}\left(t\right)=\frac{\mathrm{argmax}\left[PW_{z}\left(t,f\right)\right]}{\mu }.$$

The window length of TFPF determines its denoising ability. The denoising effect of long-window TFPF is obvious but will cause signal distortion. On the contrary, the denoising ability of short-window TFPF is poor.

### 3.4. Extreme Learning Machine

ELM is a feedforward neural network characterized by a single hidden layer. Its key innovation lies in the random generation of connection weights and biases specifically for the hidden layer, and no adjustment is needed after setting [28]. Compared to most neural networks, which need constant adjustment of their weights and thresholds, this method reduces the amount of computation by half. In addition, the output-layer connection weight β is determined once by solving the equations without iterative adjustment. Studies show that ELM has a high speed while ensuring learning accuracy [29]. The ELM network structure is given in Figure 4, and its theory is introduced as follows [30]:

For any given *N* training samples $\left\{\left(x_{1},y_{1}\right),\;\left(x_{2},y_{2}\right),\;\ldots ,\;\left(x_{N},y_{N}\right)\right\}$, where *x_i_* = [*x_i_*_1_, x*_i_*_2_,…, x*_in_*]^T^ ∈ R^n^, *y_i_* = [ *y_i_*_1_, *y_i_*_2_,…, *y_im_*]^T^ ∈ R^n^, for ELM with *k* hidden nodes, we have
(17)$$\sum_{i=1}^{k}\beta_{i}g\left(w_{i}\cdot x_{j}+b_{i}\right)=y_{j}\;j=1,\ldots ,N,$$
where *β_i_* = [*β_i_*_1_, *β_i_*_2_, …, *β_in_*]^T^ represents the output connection weights between the *i*th hidden node and the output nodes, *g*(.) represents the activation function and *w_i_* = [*w_i_*_1_, *w_i_*_2_,…, *w_in_*]^T^ represents the input connection weights.
(18)$$H=\left[\begin{array}{ccc} g\left(w_{1}•x_{1}+b_{1}\right) & \cdots & g\left(w_{k}•x_{1}+b_{k}\right)\\ \vdots & \cdots & \vdots \\ g\left(w_{1}•x_{N}+b_{1}\right) & & g\left(w_{k}•x_{N}+b_{k}\right) \end{array}\right].$$

With the introduction of matrix *H*, Equation (17) is expressed as *Hβ* = *Y*, in which *Y* = [*y*_1_^T^, *y*_2_^T^, …, *y_N_*^T^]^T^, *β* = [*β*_1_^T^, *β*_2_^T^, …, *β_k_*^T^]^T^; thus, the learning objective function for ELM can be expressed as
(19)min‖Hβ−Y‖. 

If *g*(.) is infinitely differentiable, no adjustments are necessary for the output parameters of ELM. The smallest and unique solution *β* satisfying Equation (19) can be directly calculated:(20)$$\beta =H^{†}Y,$$
where *H*^†^ is the Moore–Penrose generalized inverse of the hidden-layer output matrix *H*.

However, before using ELM, the number of neurons in the hidden layer, type of activation function, and range of input-layer connection weights need to be determined. Improper parameter settings will affect the performance of ELM, which will further affect the effect of temperature compensation.

### 3.5. Non-Dominated Sorting Genetic Algorithm II

NSGA is an improved genetic algorithm based on Pareto optimality. In this algorithm, the population is stratified according to the non-dominant relationships between individuals, and then selection, crossover, mutation and other operations are carried out. NSGA II [35] introduces the fast non-dominated sorting (FNS) method, congestion-level comparison operators, and an elite strategy to improve NSGA. This not only reduces the complexity of NSGA but also makes the Pareto frontier optimal solution evenly distributed in the whole Pareto domain, so as to alleviate the phenomenon of local optimal solutions and enhance its search ability. The following introduces FNS, the congestion comparison operator, and the elite strategy.

A.Fast non-dominated sorting.

The essence of a multi-objective optimization problem lies in obtaining the Pareto frontier optimal solution. Thus, NSGA II employs an FNS technique to explore the entire population. The population is stratified according to the non-inferiority level of individual solutions, in which the individual solutions in the next layer are all dominated by any solution in the previous layer and there is no dominant relationship between individuals in the same layer. According to this principle, the search is directed to the Pareto-optimal front.

B.Congestion comparison operator.

The crowding degree *i_d_* is defined as the density of individuals in the area where the given point is located. Firstly, the crowding degree *i_d_* of each individual is set to 0. Following FNS of the population, the crowding degree of individuals on the boundary is assigned infinity, while for others, the crowding degree is
(21)$$i_{d}=\sum_{j=1}^{m}\left|f_{j}^{i+1}-f_{j}^{i-1}\right|,$$
where *f_j_^i+^*^1^ is the *j*th objective function value of the *i*th point.

C.Elite strategy.

After FNS of the population, the crowding degrees of individuals within the generated non-dominant layers are computed. “Elite” individuals are selected based on a low non-dominant level and high crowding degree, representing the optimal solutions of the Pareto front among these elites.

Figure 5 presents the algorithmic process of NSGA II: Firstly, the initial population *P_t_*, comprising *N* individuals, is randomly generated. Subsequently, the offspring population *Q_t_* is derived using the fundamental operations of the genetic algorithm. Then, the new population *R_t_* with a population size of 2*N* is obtained by reconstructing *P_t_* and *Q_t_.* According to the elite strategy, *P_t+_*_1_ is obtained by selecting the optimal *N* individuals in *R_t_*, and their offspring population *Q_t+_*_1_ is obtained again through the genetic algorithm. The new population *R_t+_*_1_ is then constructed, and the described steps are iteratively executed until the stop condition is satisfied.

### 3.6. ELM Optimization

To enhance the generalization ability and search ability of ELM, we use NSGA II to optimize the number of neurons in the hidden layer, the type of activation function, and the range of input-layer connection weights. The key technology of NSGA II-ELM lies in the following.

(1)The definition of population individuals.

The mapping relationship between the parameters to be optimized in ELM and the population individuals of the NSGA II algorithm is established. Here, the optimization interval of the hidden neurons was set to [0, 50], and the five most commonly used activation functions—‘sigmoid’, ‘sin’, ‘hardlim’, ‘tribas’ and ‘radbas’—were selected for testing. In order to optimize the input-layer connection weights, the scaling factor λ is introduced to limit the range of the input-layer connection weights. Therefore, the value range of the input-layer connection weights can be expressed as
(22)$$w_{ij}\in \lambda \cdot [-1,1]\;\lambda \in [0,1],$$
where *w_ij_* represents the input connection weights, and the randomly set range of the input-layer weights will change with the value of the scaling factor λ, so as to achieve the purpose of limiting their range.

(2)The determination of the objective function.

For ELM, the smaller the output weights and prediction error, the better the network generalization ability. Therefore, the RMSE between the predicted and true values and the 2-norm of the output weight were taken as the two objective functions in this paper.
(23)$$\left\{\begin{array}{l} f_{1}=\min\sqrt{\frac{\sum_{i=1}^{N}{\left(y_{p,i}-y_{t,i}\right)}^{2}}{N}},\\ f_{2}=\min\|\beta {\|}_{2}^{2} \end{array}\right.$$
Here, *N* is the number of input samples, and *y_p,i_* and *y_t,i_* are the predicted value and true value, respectively, of the *i*th training sample.

### 3.7. ICEEMDAN-SE-TFPF and NSGA II-ELM

Relying on the aforementioned algorithms, we propose a hybrid algorithm named ICEEMDAN-SE-TFPF and NSGA II-ELM for gyroscope temperature compensation. The algorithm logic diagram is shown in Figure 6, and the specific steps are as follows:

Step 1. Firstly, the gyroscope’s output containing noise and temperature drift is decomposed into IMFs by ICEEMDAN.

Step 2. Secondly, the decomposed IMFs are classified into a noise segment, mixed segment and feature segment by SE.

Step 3. Long-window TFPF is used to denoise the noise segment with high noise content, short-window TFPF is used to strip the useful signals in the mixed segment, and the feature segment containing temperature drift is compensated by NSGA II-ELM.

Step 4. Finally, the denoised noise segment, denoised mixed segment and compensated feature segment are combined to reconstruct the final gyroscope’s output.

## 4. Experiment

### 4.1. The Experimental Process

The setup for the temperature experiment on the gyroscope is shown in Figure 7. A detection circuit was distributed across three separate PCBs. Metal pins served not only to interconnect the electronic signals within the detection circuit but also to facilitate the connections between the three PCBs, each of which was encased in a rubber pad. Subsequently, the rubber-wrapped PCBs were placed within a metal shell, a measure that effectively safeguarded the chip’s structure while minimizing the risk of severe impact. To mitigate electromagnetic interference, the ground signal was connected to the metal shell. Additionally, one of the three PCBs served as a weak signal interface with the structural chip, while the other two functioned as induction and drive circuits, respectively. The experimental setup included a temperature-controlled oven, a multimeter, a signal generator, and a DC power supply of ±10 V. The experimental procedure unfolded as follows: Initially, activate the gyroscope and allow it to stabilize at room temperature for one hour. Next, rapidly raise the temperature-controlled oven to 60 °C to ensure that the gyroscope housing reaches a consistent temperature of 60 °C, maintaining this temperature for an additional hour. Subsequently, as the temperature in the oven decreases to 10 °C, the gyroscope continues to operate under these conditions for an hour. Finally, as the temperature drops to −40 °C, the gyroscope operates for one final hour before concluding the experiment.

### 4.2. The Experimental Results

Figure 8 shows the temperature experiment results from the gyroscope. It can be seen that the gyroscope’s output presents a nonlinear relationship with the temperature change. As the temperature changed from 60 °C to −40 °C, the output shifted from 0.115° to 0.15°, and the gyroscope’s output signal contains not only temperature drift but also noise.

According to the flow of our proposed hybrid algorithm, the gyroscope’s output signal was firstly decomposed by ICEEMDAN. Figure 9 shows that the gyroscope’s output signal was decomposed into 12 IMFs. The SE value of each IMF was calculated and classified, and the results are shown in Figure 10. The SE values of the first, second and third IMFs were greater than 0.6, so these IMFs were divided into the noise segment, which contains a lot of noise. Therefore, the long-window TFPF with better denoising ability was selected for denoising this segment. The SE values of the fourth, fifth, sixth and seventh IMFs were between 0.1 and 0.6, so these IMFs were divided into the mixed segment containing noise and useful signals. Because short-window TFPF can well preserve useful components during denoising, it was selected for denoising this segment. The SE values of the 8th, 9th, 10th, 11th and 12th IMFs were below 0.1, so these IMFs were divided into the feature segment, which comprises a drift term caused by the temperature change. This segment was compensated by NSGA II-ELM.

Before establishing the temperature compensation model to compensate the feature segment, NSGA II was used to optimize ELM to obtain the best network parameters. As mentioned before, the optimal searching range for hidden neurons was [0, 50]; the activation functions were ‘sigmoid’, ‘sin’, ‘hardlim’, ‘tribas’ and ‘radbas’; and the range of connection weights of the input layer was set to [−1, 1]. In addition, the population number was set to 40, the crossover probability was 0.8, the mutation probability was 0.15, and the number of iterations was 15. After the iterations, the optimization result of the NSGA II particle swarm was obtained. Figure 11 shows the distribution of particles in three-dimensional space, where each blue circle represents a particle. The position of the particle marked by the red circle represents the optimal network parameters for ELM, that is, the number of hidden neural networks was 11, the activation function type was the ‘sin’ function, and the range of input-layer connection weights was [−0.65, 0.65]. Figure 12 presents a comparison of each segment before and after processing. In order to investigate the denoising capability of ICEEMDAN-SE-TFPF, we reconstructed the noise segment after denoising, the mixed segment after denoising, and the feature segment. From Figure 13, we can see that the noise was basically eliminated. On this basis, the final signal was obtained by compensating the feature segment. In Figure 14, we can see that the noise component and temperature drift in the gyroscope’s output signal were eliminated. To further evaluate the proposed algorithm, a quantitative analysis was performed, as described in the next subsection.

### 4.3. Comparative Analysis

Inertial devices commonly employ Allan variance analysis, as outlined in the IEEE standard [36], for error analysis. In this study, we utilized Allan variance analysis to quantitatively assess the angle random walk and bias stability of both the original and compensated gyroscope outputs. Figure 15 illustrates the comparison results. Following denoising and compensation, the angle random walk of the original output decreased from 0.531076 to 6.65894 × 10^−3^°/h/√Hz, while the bias stability was reduced from 32.7364°/h to 0.259247°/h. These findings underscore the superior performance of our proposed hybrid algorithm. A high-precision model is key to improving the compensation effect, so we adopted NSGA II to optimize ELM. To verify the superiority of the improved ELM, a comparative analysis was performed here with a BP neural network and unoptimized ELM, taking the experimental temperature change as the input and the drift part in the gyroscope output as the output for training. In Figure 16, the error of ELM at the inflection point is large when compared with the real output, and the predicted output of the improved ELM is closest to the real value when compared with BP. After the input parameters of ELM were optimized by NSGA II, ELM had stronger generalization ability and learning accuracy.

## 5. Conclusions

In this paper, we proposed a hybrid algorithm based on ICEEMDAN-SE-TFFPF and NSGA II-ELM for gyroscope temperature compensation. After ICEEMDAN decomposition and SE classification, the gyroscope’s output signal was divided into a noise segment, mixed segment and feature segment; then, long-window and short-window TFPF were selected to denoise the noise segment and the mixed segment, respectively. Before processing the feature segment, NSGA II was utilized to optimize ELM, including the number of hidden-layer neurons, the type of activation function, and the range of input-layer connection weights. The optimization objectives were determined by the RMSE of the predicted output and the 2-norm of the output-layer connection weight. This gave ELM good generalization ability and higher prediction accuracy; when compared with the BP neural network and ELM, the predicted value of improved ELM was closest to the real value, displaying improved compensation accuracy. The experimental results show that the angle random walk and bias stability of the compensated output signal were reduced from 0.531076 to 6.65894 × 10^−3^°/h/√Hz and from 32.7364 to 0.259247°/h, respectively.

## Figures and Tables

**Figure 1 micromachines-15-00609-f001:**
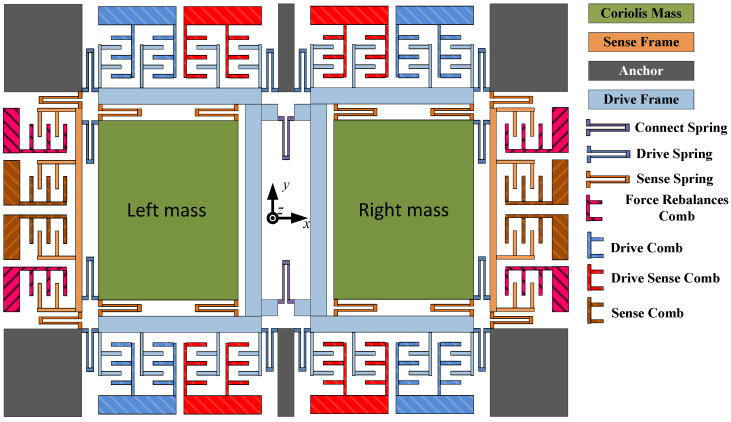
Structure of the dual-mass MEMS gyroscope.

**Figure 2 micromachines-15-00609-f002:**
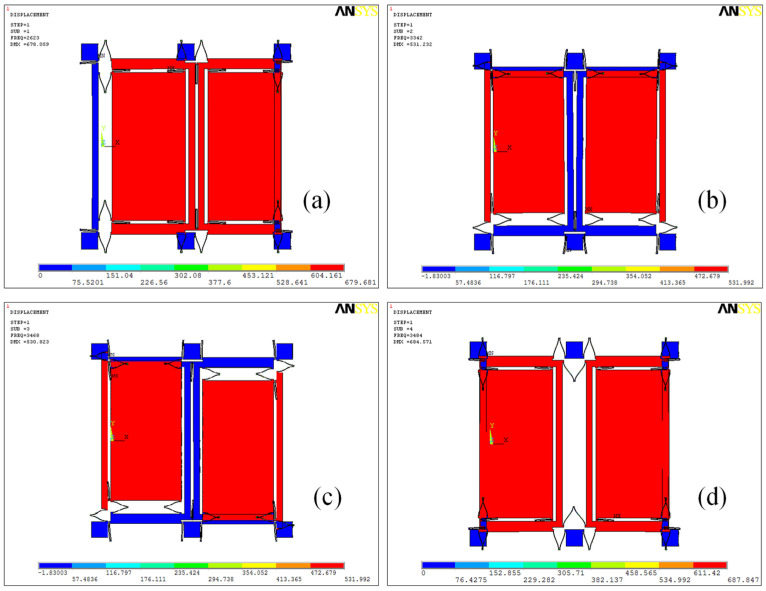
The gyroscope’s four operational modes. Subfigures (**a**–**d**) represent each mode, with their corresponding frequencies denoted as *ω*_1_ = 2623 × 2π rad/s, *ω*_2_ = 3342 × 2π rad/s, *ω*_3_ = 3468 × 2π rad/s and *ω*_4_ = 3484 × 2π rad/s.

**Figure 3 micromachines-15-00609-f003:**
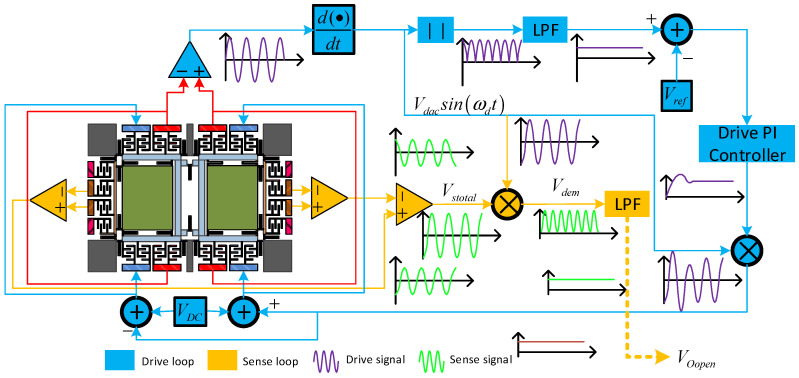
A description of the gyroscope’s monitoring system.

**Figure 4 micromachines-15-00609-f004:**
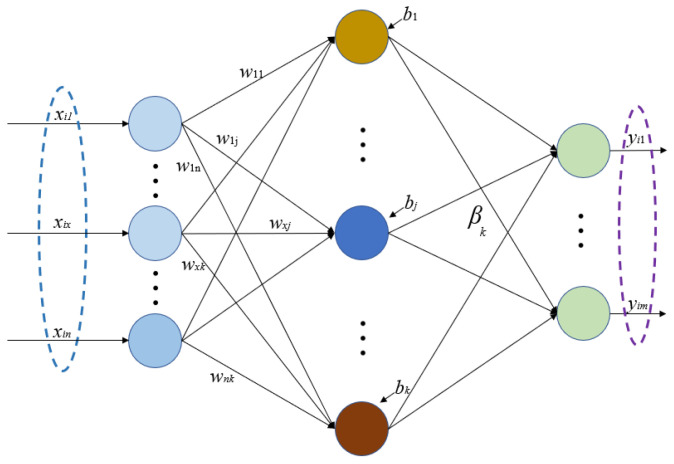
The network structure of ELM.

**Figure 5 micromachines-15-00609-f005:**
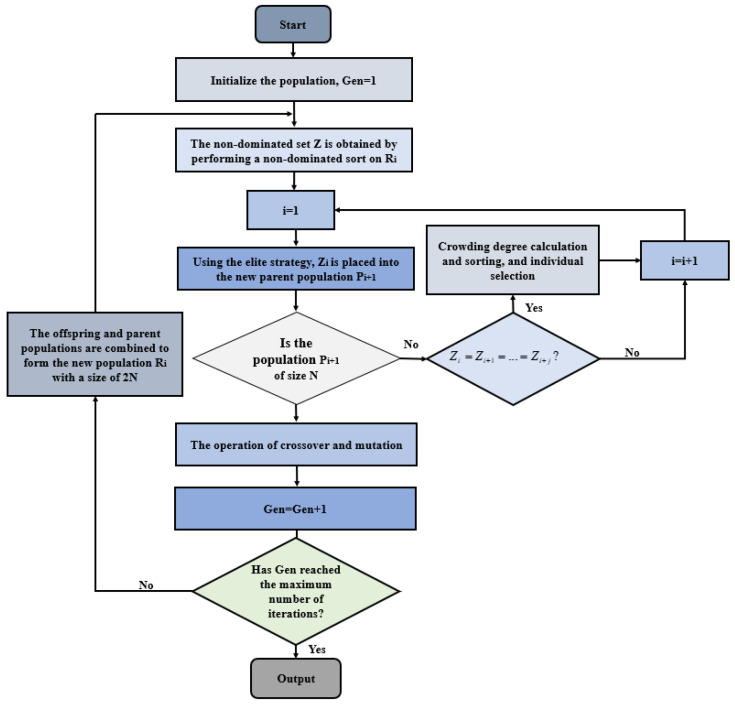
The algorithm process of NSGA II.

**Figure 6 micromachines-15-00609-f006:**
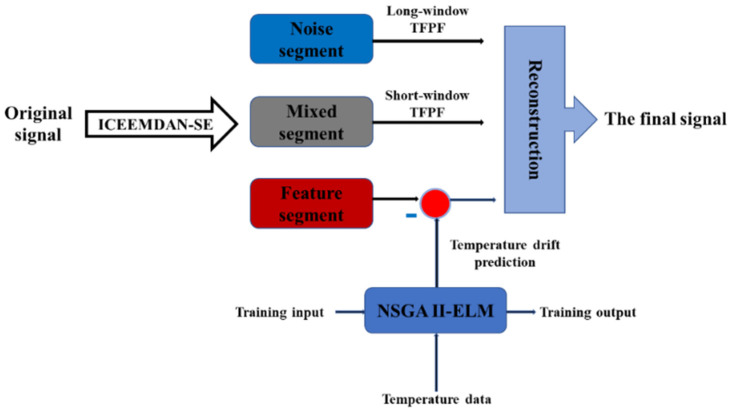
The framework of ICEEMDAN-SE-TFPF and NSGA II-ELM.

**Figure 7 micromachines-15-00609-f007:**
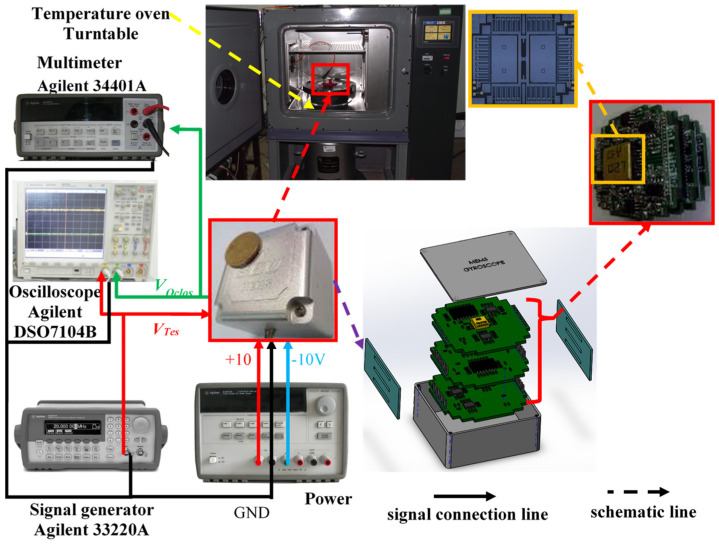
Gyroscope temperature test equipment.

**Figure 8 micromachines-15-00609-f008:**
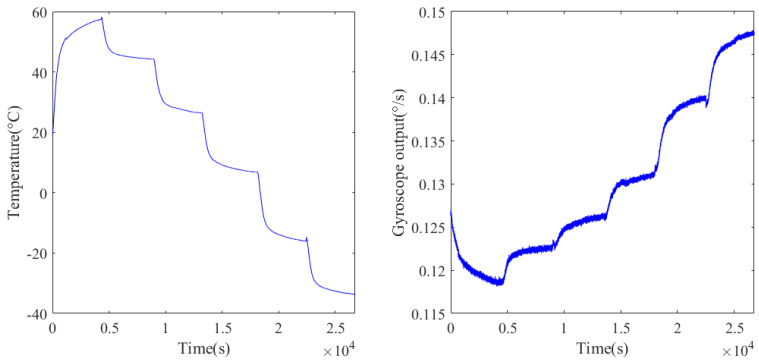
Experimental temperature change and the corresponding gyroscope output.

**Figure 9 micromachines-15-00609-f009:**
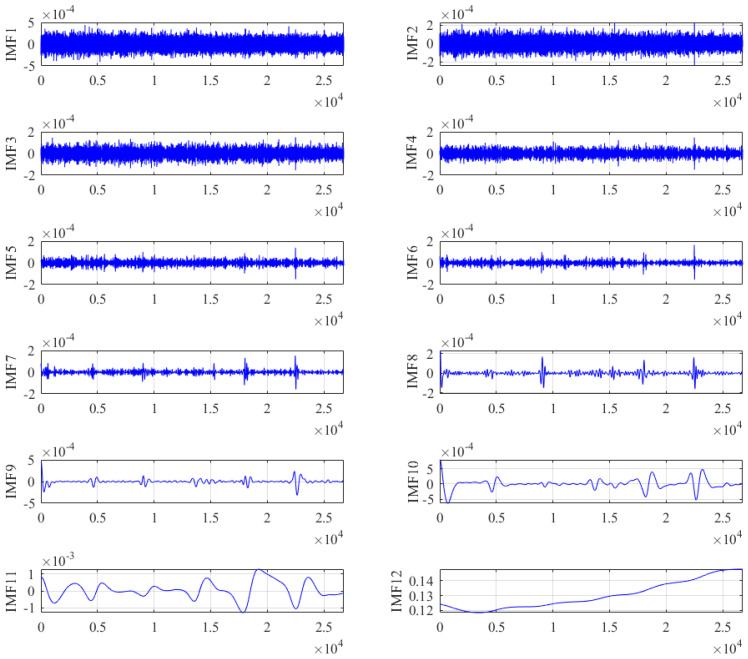
Decomposition results of the gyroscope’s output signal based on ICEEMDAN.

**Figure 10 micromachines-15-00609-f010:**
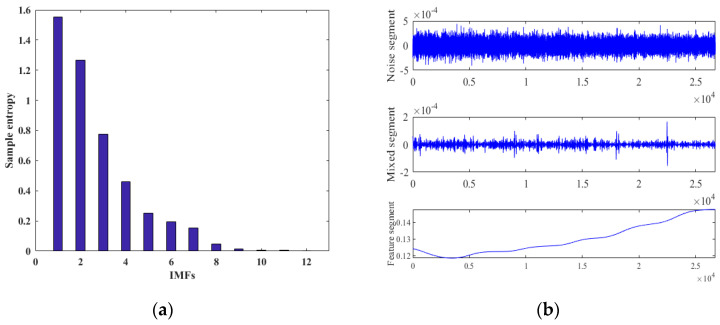
Classification results. (**a**) SE value of each IMF. (**b**) Examples of different segments.

**Figure 11 micromachines-15-00609-f011:**
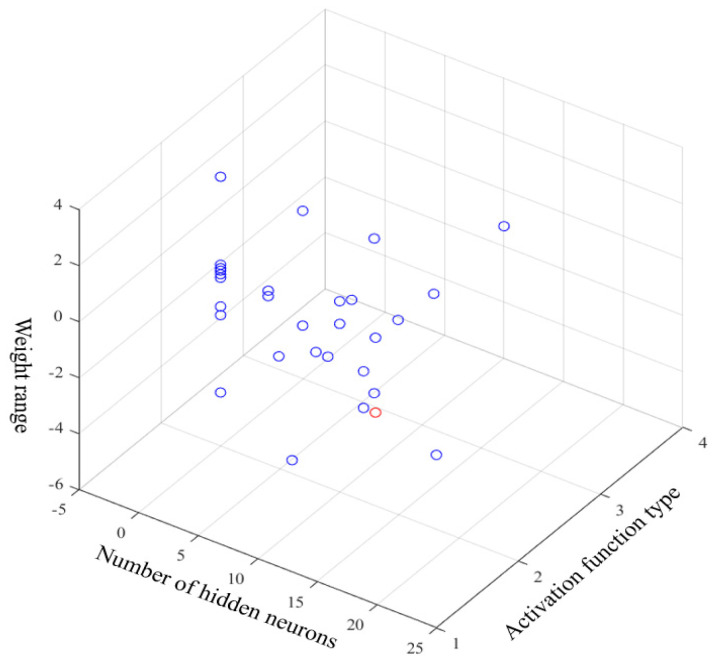
Distribution of particles in the NSGA II search.

**Figure 12 micromachines-15-00609-f012:**
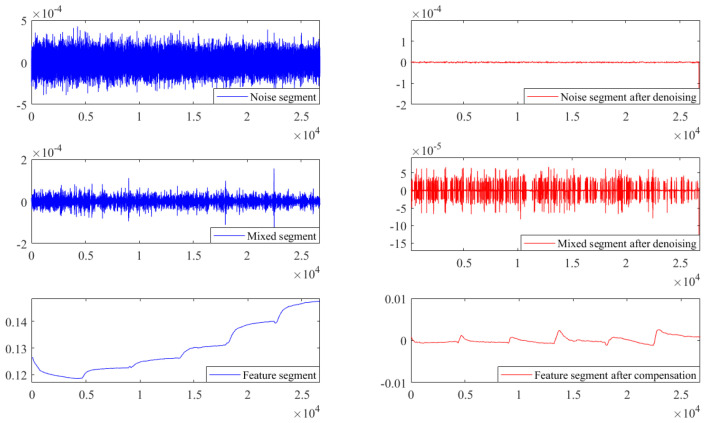
Noise reduction or compensation results for each segment.

**Figure 13 micromachines-15-00609-f013:**
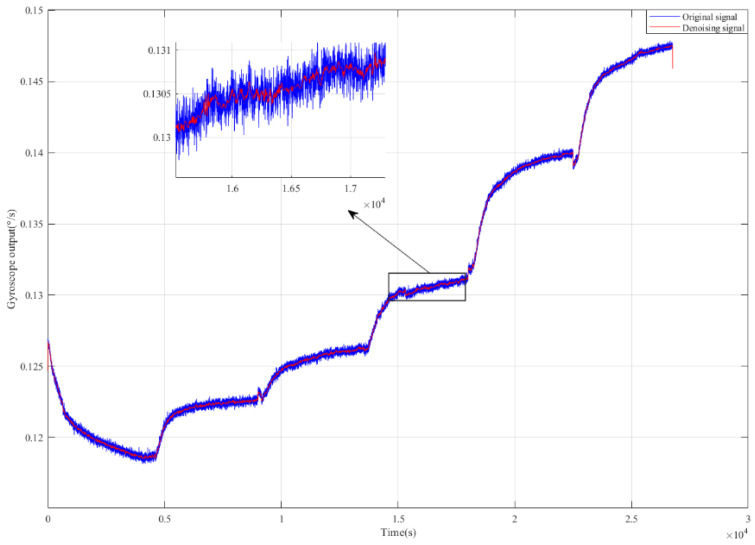
A comparison between the gyroscope’s outputs before and after denoising.

**Figure 14 micromachines-15-00609-f014:**
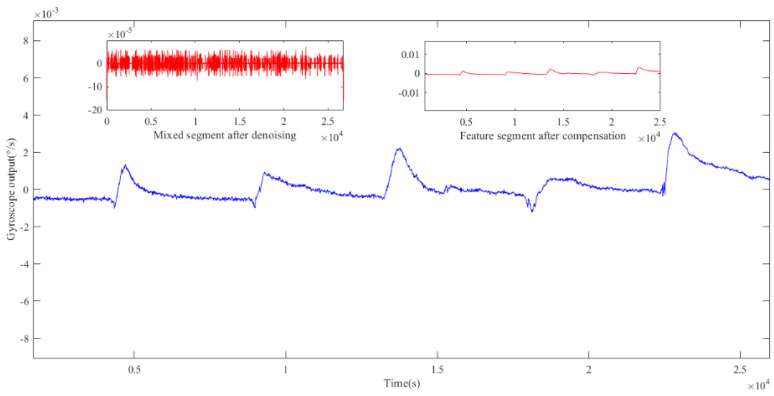
Final output from the gyroscope based on ICEEMDAN-SE-TFPF and NSGA II-ELM.

**Figure 15 micromachines-15-00609-f015:**
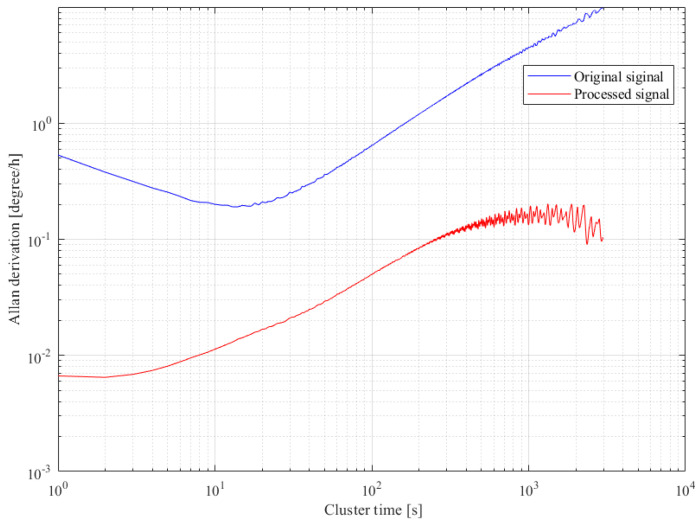
Allan variance analysis for output signals.

**Figure 16 micromachines-15-00609-f016:**
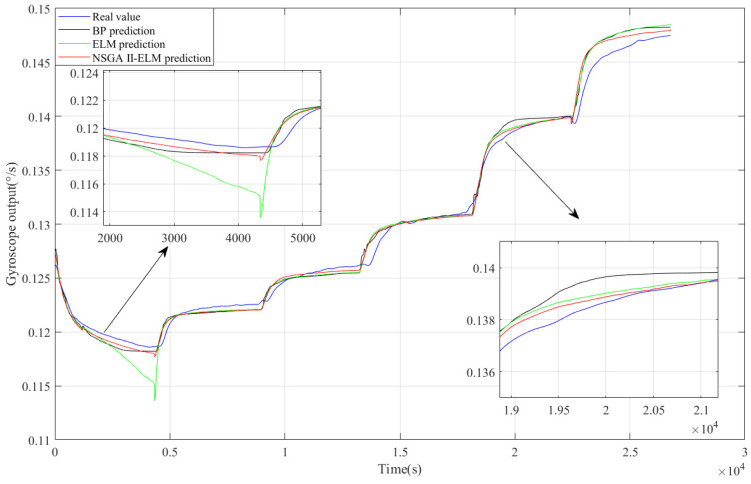
Comparison result for the BP neural network, ELM and NSGA II-ELM.

## Data Availability

Data are contained within the article.

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
