# Peer review of "MEMS Gyroscope Temperature Compensation Based on Improved Complete Ensemble Empirical Mode Decomposition and Optimized Extreme Learning Machine"

_micromachines, 2024, doi:10.3390/mi15050609_

Round 1

Reviewer 1 Report

Comments and Suggestions for Authors

The manuscript presents a comprehensive overview of a hybrid algorithm for MEMS gyroscope temperature compensation with ICEEMDAN-SE-TFPF and NSGAâ…¡-ELMs. It demonstrate the Components and implementation process of of the hybrid algorithm for MEMS gyroscope temperature compensation, which is verified by experiments. The experimental results show that the algorithm can improve the angular random walk and bias stability of gyroscope by about 100 times respectively. Overall, the manuscript is well-written, clear, and easy to understand, making it a valuable resource for students and researchers in the field of MEMS gyroscope temperature compensation. But there are still some parts that are not understood or need improvement. Please help answer and modify them, specifically:

1. In line 351, what are the characteristics and physical significance of the "feature segment" mentioned?

2. In the paragraphs corresponding to lines 389 to 399, it is mentioned that IMFs are divided into "noise segment", "mixed segment" and "feature segment" according to their values. Why are they divided according to these values? Is there any basis?

3. In Figure 14, there is still fluctuation in the output after compensation. What is the cause?

Author Response

Dear reviewer, we have revised our manuscript according to your suggestion, please see the attachment for details.

Reviewer 2 Report

Comments and Suggestions for Authors

Nice work which can be published. I have only two notes.

1. The introduction has to be expanded. The authors are to outline in more details the device under the study and the problems which are to be fixed. All abbreviations are to be explained and maybe some of them are to be replaced by the full form - to the moment there are too many abbreviations and hence the reading is tantalized. All the methods, which are used in the design and study, are to be not only mentioned but also briefly introduced and explained.

2. To our opinion it is absolute inappropriate to use abbreviations in the paper title - it has to be clkear even for those readers, who do not know what is ICEEMDAN-SE-TFPF or NSGAâ…¡-ELM 

Comments on the Quality of English Language

The language of the manuscript is understandable but requires editing by somebody with a fluent scientific English.

Author Response

(The authors gave the same response as above.)
